# ReText: Text Boosts Generalization in Image-Based Person Re-identification

## Abstract

Generalizable image-based person re-identification (Re-ID) aims to recognize individuals across cameras in unseen domains without retraining. While multiple existing approaches address the domain gap through complex architectures, recent findings indicate that better generalization can be achieved by stylistically diverse single-camera data. Although this data is easy to collect, it lacks complexity due to minimal cross-view variation. We propose ReText, a novel method trained on a mixture of multi-camera Re-ID data and single-camera data, where the latter is complemented by textual descriptions to enrich semantic cues. During training, ReText jointly optimizes three tasks: (1) Re-ID on multi-camera data, (2) image-text matching, and (3) image reconstruction guided by text on single-camera data. Experiments demonstrate that ReText achieves strong generalization and significantly outperforms state-of-the-art methods on cross-domain Re-ID benchmarks. To the best of our knowledge, this is the first work to explore multimodal joint learning on a mixture of multi-camera and single-camera data in image-based person Re-ID. Code will be made publicly available.

## 1 Introduction

Image-based person re-identification (Re-ID) is the task of recognizing individuals across non-overlapping cameras. While modern Re-ID methods achieve strong performance in single-domain scenarios, their effectiveness often drops significantly when applied to unseen domains without retraining. This setting, known as generalizable person Re-ID, is essential for real-world applications, where collecting labeled data for each new environment is infeasible.

Achieving generalization in Re-ID is particularly challenging due to the lack of large-scale and diverse multi-camera data. Such datasets are expensive and labor-intensive to collect, as they require synchronized camera networks and identity annotations across views. As a result, Re-ID models tend to overfit to the training domain and fail to generalize to new domains with different visual and view distributions.

Recent works attempt to address the domain gap by designing increasingly complex architectures (Liao & Shao, 2020; 2021) and training strategies (Ni et al., 2023; Cho et al., 2024). Another line of research explores single-camera data, which is stylistically diverse and easy to collect. However, it lacks the cross-view variation central to Re-ID, limiting its effectiveness when used alone (Sec. 4.3). ReMix (Mamedov et al., 2025) shows that combining multi-camera and single-camera data improves generalization but does not fully exploit the semantic potential of the latter and leaves its complexity unexplored.

In parallel, methods like CLIP-ReID (Li et al., 2023) investigate textual supervision for person Re-ID by introducing language as an additional modality. However, they do not exploit single-camera data, and multi-camera datasets lack descriptive captions, so these methods typically rely on learnable text tokens. As a result, the semantic richness and generalization capability of natural language remain underutilized in current approaches.

In this work, we propose ReText, a novel method trained on a mixture of multi-camera Re-ID data and single-camera data with textual descriptions. Our key insight is that while single-camera data is less complex due to limited cross-view variation for person Re-ID, pairing it with descriptive captions unlocks rich semantic cues that significantly enhance domain generalization (Fig. 1). ReText

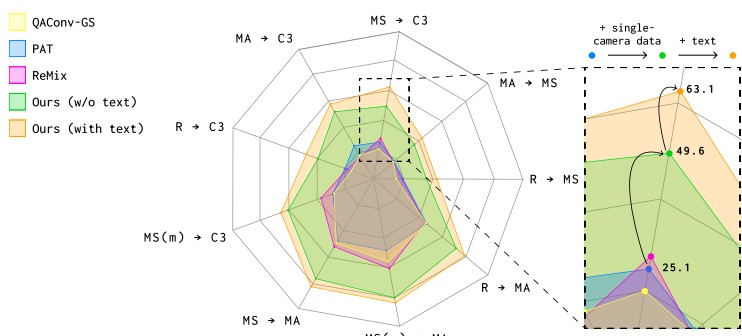

Figure 1: Impact of single-camera data and textual supervision on cross-domain Re-ID performance ($mAP$). Adding single-camera data improves generalization (Ours w/o text) and further gains are achieved by incorporating textual descriptions (Ours with text), highlighting the effectiveness of multimodal supervision during training.

thus addresses two major challenges simultaneously: the scarcity of multi-camera data, which we mitigate by leveraging stylistically diverse single-camera data, and the underutilization of natural language supervision.

During training, ReText jointly optimizes three tasks: (1) Re-ID on multi-camera data, (2) image-text matching, and (3) image reconstruction guided by text on single-camera data. Since semantic cues provided by natural language — such as clothing, attributes, and actions — are often invariant across domains, this design encourages the model to learn domain-agnostic representations. Unlike prior approaches, ReText is the first image-based person Re-ID method to study the complexity of single-camera data while also incorporating descriptive captions instead of learnable text tokens. As a result, it can better utilize both the single-camera data and the semantic structure of language only during training to improve generalization.

Our main contributions are summarized as follows:

- We present ReText, a novel method that jointly leverages multi-camera Re-ID data and single-camera data with textual descriptions.
- We design a three-task training strategy: Re-ID for identity-discriminative learning, image-text matching for cross-modal alignment, and image reconstruction guided by text to enhance robustness to missing or occluded visual information.
- We introduce Identity-aware Matching and Structure-preserving losses that explicitly address the specifics of Re-ID, ensuring both cross-modal alignment and intra-class consistency for effective image-text matching.

Our experiments demonstrate that ReText significantly outperforms state-of-the-art methods across all cross-domain Re-ID benchmarks.

## 2 RELATED WORK

### 2.1 GENERALIZABLE PERSON RE-ID

Generalizable person Re-ID focuses on training models that perform well in unseen domains without retraining. Prior works in this area have explored various strategies to improve cross-domain generalization, including architecture-level modifications (Liao & Shao, 2020; 2021), normalization techniques (Jiao et al., 2022; Qi et al., 2022; Zhang et al., 2023), and meta-learning approaches (Choi et al., 2021; Zhao et al., 2021). Other methods have concentrated on domain-invariant feature learning (Zhang et al., 2022; Lin et al., 2023) and part-aware modeling (Ni et al., 2023). In addition to architectural and training innovations, several recent works have proposed more effective mini-batch sampling approaches to encourage domain robustness during training (Liao & Shao, 2022; Zhao et al., 2024).

Despite significant progress in model architectures and training strategies, the role of training data diversity and language-based supervision remains underexplored in generalizable person Re-ID. ReText highlights the importance of both: we demonstrate that joint training on a mixture of multi-camera Re-ID data and stylistically diverse single-camera data with textual descriptions significantly boosts domain generalization and achieves state-of-the-art performance across multiple cross-domain Re-ID benchmarks.

## 2.2 USE OF SINGLE-CAMERA DATA

Single-camera data offers a promising source of training samples due to its abundance and stylistic diversity. It can be collected at scale through automated processing of publicly available video streams, such as YouTube videos (Fu et al., 2021; 2022). However, from the perspective of the person Re-ID task, single-camera data is inherently simpler, as it lacks cross-view variation — a core challenge in Re-ID. Consequently, it has primarily been used for self-supervised pre-training, followed by fine-tuning on multi-camera Re-ID datasets (Mamedov et al., 2023; Hu et al., 2024). In contrast, ReMix (Mamedov et al., 2025) identified a bottleneck in this two-stage process and proposed a joint training procedure that combines both multi-camera and single-camera data. This approach significantly improved generalization, demonstrating that single-camera data can be a valuable asset when incorporated correctly.

Yet, ReMix does not explicitly account for the simplicity of single-camera data. ReText introduces a fundamentally different strategy by using single-camera data not only as a source of stylistic diversity but also as language-supervised data. By pairing such data with descriptive captions, semantic complexity is introduced, which compensates for the lack of cross-view variation and enhances generalization.

## 2.3 MULTIMODAL LEARNING

Multimodal models such as CLIP (Radford et al., 2021), ALBEF (Li et al., 2021), and BLIP (Li et al., 2022) have demonstrated strong cross-modal learning capabilities. Building on these foundations, recent works have introduced multimodal learning into person Re-ID (Li et al., 2023; Zhao et al., 2024; 2025). Such methods optimize learnable text tokens to encode identity-specific semantics, and fine-tune the image encoder using these tokens through the CLIP loss.

Since the CLIP loss treats images of the same identity as different classes, it is suboptimal for person Re-ID, where such images are often similar. Our proposed Identity-aware Matching and Structure-preserving losses overcome this drawback, while the Reconstruction Loss further improves robustness to missing or occluded visual information.

## 2.4 DIFFERENCE FROM EXISTING MULTIMODAL PERSON RE-ID METHODS

Existing multimodal Re-ID methods, such as Instruct-ReID (He et al., 2024) and Instruct-ReID++ (He et al., 2025) aim to handle multiple Re-ID scenarios within a single model: image-based, text-based, visible–infrared, clothes-changing, etc. In contrast, ReText focuses solely on image-based person Re-ID and introduces a different multimodal setting: joint training on multi-camera data and single-camera data enriched with textual descriptions.

Prior works neither use text to increase the complexity of single-camera data nor study how this affects cross-dataset generalization. Our three-task learning strategy and new loss functions are tailored specifically to improve visual representations for image-based person Re-ID, making ReText complementary to existing multimodal approaches rather than overlapping with them.

## 3 PROPOSED METHOD

ReText consists of four core components: an image encoder, a momentum image encoder, a text encoder, and an image decoder (Fig. 2). During inference, only the momentum image encoder is used, meaning the inference is image-only. Such image encoder modification is widely used in recent person Re-ID methods, such as ReMix, which updates via exponential moving average and serves to stabilize training, particularly for multimodal objectives.

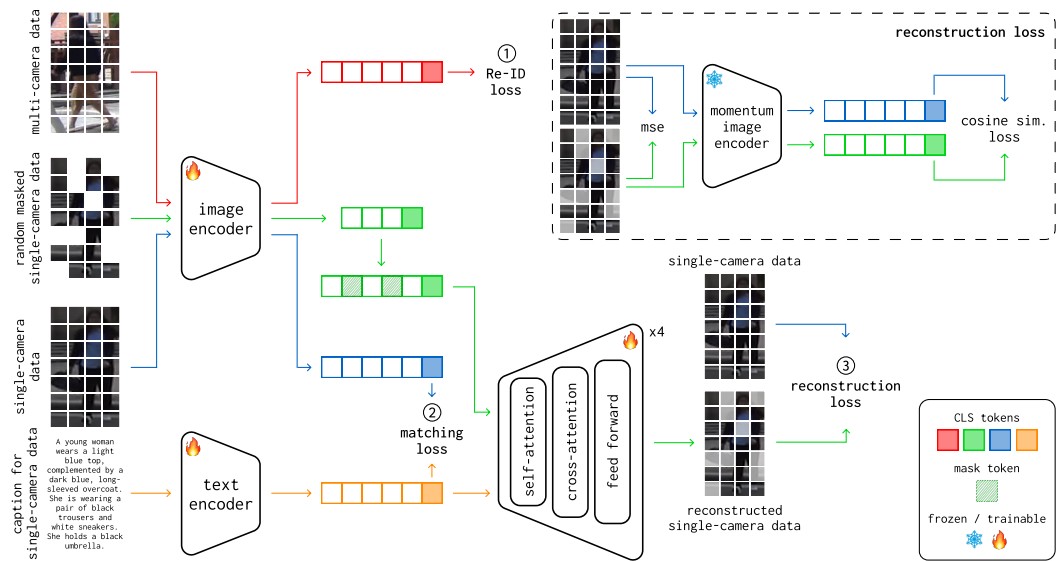

Figure 2: Overview of ReText. The model receives four types of inputs: multi-camera data, single-camera data, masked single-camera data, and textual captions. It jointly optimizes three loss functions: Re-ID loss on multi-camera data, image-text matching loss on single-camera data, and text-guided reconstruction loss on masked single-camera data.

During training, ReText receives four types of inputs: multi-camera data, single-camera data, masked single-camera data, and corresponding textual captions. Using this data, the model jointly optimizes the following three tasks: (1) Re-ID task on multi-camera data (Sec. 3.2), (2) image-text matching task on single-camera data and their corresponding textual captions (Sec. 3.3), and (3) image reconstruction task on masked single-camera data guided by textual captions (Sec. 3.4).

All three tasks contribute equally to the final objective, and their combination allows ReText to learn discriminative, semantically grounded, and domain-invariant representations.

## 3.1 FORMAL DEFINITIONS

Let $\{(x_i, t_i, y_i)\}_{i=1}^{N}$ denote a batch of single-camera images $x_i$, their corresponding textual captions $t_i$, and identity labels $y_i$. The image encoder $f_\theta$ and text encoder $g_\phi$ are both transformer-based. For both modalities, the $[CLS]$ token is used as a global representation:

$$h_i^{img} = CLS(f_\theta(x_i)), \ h_i^{txt} = CLS(g_\phi(t_i)). \tag{1}$$

These modality-specific embeddings are then projected into a shared embedding space using linear projection heads:

$$z_i^{img} = W_{img}h_i^{img}, \ z_i^{txt} = W_{txt}h_i^{txt}. \tag{2}$$

Hereafter, $\hat{z}_i^{img}$ and $\hat{z}_i^{txt}$ denote the $\ell_2$-normalized versions of $z_i^{img}$ and $z_i^{txt}$, respectively.

## 3.2 RE-ID TASK

The Re-ID task focuses on learning identity-discriminative features across different camera views. To this end, ReText adopts the loss formulation from ReMix, which combines four components: the Instance ($\mathcal{L}_{ins}$), Augmentation ($\mathcal{L}_{aug}$), Centroids ($\mathcal{L}_{cen}$), and Camera Centroids ($\mathcal{L}_{cc}$) losses. The total Re-ID loss is defined as:

$$\mathcal{L}_{ReID} = \mathcal{L}_{ins} + \mathcal{L}_{aug} + \mathcal{L}_{cen} + 0.5 \cdot \mathcal{L}_{cc}. \tag{3}$$

The Instance Loss pulls anchors closer to positive instances while distancing them from negative ones, promoting a more generalized model solution. The Augmentation Loss aligns augmented images with their originals and separates them from other identities, addressing inter-instance similarity changes from augmentations. The Centroids Loss brings instances closer to their corresponding

centroids and pushes them away from other centroids. The Camera Centroids Loss groups instances with the same label but captured by different cameras around their respective centroids.

Unlike ReMix, which applies these losses to both multi-camera and single-camera data, ReText uses them only for multi-camera data. This design choice is motivated by the fact that Re-ID supervision is most effective when cross-view variation is present (Sec. A.2). Single-camera data, by contrast, is considerably simpler and less informative for direct cross-view identity classification (Sec. 4.3).

### 3.3 IMAGE-TEXT MATCHING TASK

To exploit the semantic information provided by natural language, ReText introduces the image-text matching task applied to single-camera data and their corresponding textual descriptions. This task encourages the model to align visual and textual modalities, thereby learning semantically meaningful and domain-invariant representations.

For this task, we propose two new loss functions that explicitly address the specifics of Re-ID:

- The Identity-aware Matching Loss ($\mathcal{L}_{im}$) — unlike one-to-one matching in CLIP, this loss learns a distribution over multiple positive image-text pairs, enabling more flexible and identity-aware alignment.
- The Structure-preserving Loss ($\mathcal{L}_{sp}$) — enhances the structure of image representations in the shared embedding space.

These two losses operate in the same embedding space and are jointly optimized to encourage cross-modal alignment while preserving the intra-class structure among image embeddings. The total image-text matching loss is defined as:

$$\mathcal{L}_{match} = \mathcal{L}_{im} + \mathcal{L}_{sp}. \tag{4}$$

### 3.3.1 THE IDENTITY-AWARE MATCHING LOSS

To better capture identity-aware alignment between images and texts, ReText formulates the matching objective as a divergence between the predicted similarity distribution and a soft target distribution over all positive image-text pairs.

Given a projected image embedding $z_i^{img}$ and a set of normalized projected text embeddings $\{\hat{z}_j^{txt}\}_{j=1}^N$, the similarity-based probability distribution is computed as:

$$p_{i,j} = \frac{\exp\left((z_i^{img})^\top \hat{z}_j^{txt}\right)}{\sum_{k=1}^N \exp\left((z_i^{img})^\top \hat{z}_k^{txt}\right)}. \tag{5}$$

The corresponding soft target distribution $q_{i,j}$ is defined based on identity labels:

$$q_{i,j} = \begin{cases} \alpha, & \text{if } i = j \\ \frac{1-\alpha}{N_{pos}}, & \text{if } y_i = y_j \text{ and } i \neq j \\ 0, & \text{otherwise} \end{cases}, \tag{6}$$

where $N_{pos}$ is the number of positive instances in the mini-batch with identity $y_i$, and $\alpha \in (0, 1)$ controls the relative weight of the main image-text pair versus other positives.

The image-to-text component of the loss is defined as a KL-divergence between these two distributions:

$$\mathcal{L}_{it} = \frac{1}{N} \sum_{i=1}^N \sum_{j=1}^N p_{i,j} \log \frac{p_{i,j}}{q_{i,j} + \epsilon}, \tag{7}$$

where $\epsilon$ is a small constant for numerical stability.

The text-to-image component $\mathcal{L}_{ti}$ is computed in the same way, by swapping the roles of image and text embeddings. In this way, the final Identity-aware Matching Loss combines both directions:

$$\mathcal{L}_{im} = \mathcal{L}_{it} + \mathcal{L}_{ti}. \tag{8}$$

Compared to CLIP-style contrastive loss, which performs strict one-to-one matching, our loss learns a distribution over multiple positive pairs. It assigns relative weight to both the main image-text pair and other same-identity instances, enabling more flexible and semantically grounded supervision. This structure-aware formulation improves generalization by capturing richer intra-class variation (Sec. 4.4.2).

### 3.3.2 THE STRUCTURE-PRESERVING LOSS

While the Identity-aware Matching Loss aligns image and text embeddings across modalities, it does not explicitly enforce structural consistency among image embeddings of the same identity. To address this, we introduce the Structure-preserving Loss, which encourages the preservation of intra-class structure within the shared embedding space.

The loss focuses on the hardest positive instance for each anchor and is defined as:

$$\mathcal{L}_{sp} = \mathbb{E}\left[ -\log \frac{\exp\left(\langle \hat{z}_i^{img} \cdot \hat{z}_j^{img}\rangle/\tau\right)}{\sum_{k=1}^{K} \exp\left(\langle \hat{z}_i^{img} \cdot \hat{z}_k^{img}\rangle/\tau\right)} \right], \tag{9}$$

where $j = \arg\min_{j=1,..,N_{pos}} \left( \langle \hat{z}_i^{img} \cdot \hat{z}_j^{img}\rangle \right)$ is the index of the hardest positive pair for anchor $i$, $K = N - N_{pos} - 1$ is the number of negative instances that do not belong to the identity $y_i$ in the mini-batch, and $\tau$ is a temperature hyperparameter.

### 3.4 IMAGE RECONSTRUCTION TASK

To further align visual and textual modalities and enrich semantic understanding, ReText introduces the text-guided image reconstruction task. The objective is to reconstruct the masked patches of an input image by leveraging both the visible visual tokens and the corresponding textual description. This task reinforces the ability of the model to ground semantic cues in visual content and enhances its robustness to missing or occluded information.

### 3.4.1 ARCHITECTURE

Given an image $x_i$ from the single-camera data, randomly selected patches are removed and a masked image $\tilde{x}_i$ is obtained. The visible (unmasked) patches are passed through the image encoder $f_\theta$, while the masked positions are replaced with learnable mask tokens. In parallel, the corresponding textual caption $t_i$ is encoded via the text encoder $g_\phi$. Both representations are then fed into the decoder $d_\psi$, which predicts the pixel values for the missing patches.

### 3.4.2 THE RECONSTRUCTION LOSS

ReText employs two complementary loss functions to supervise the image reconstruction process. The MSE loss encourages accurate recovery of the original image patches:

$$\mathcal{L}_{mse} = \frac{1}{M} \sum_{m=1}^{M} \left\| x_i^{(m)} - \hat{x}_i^{(m)} \right\|_2^2, \tag{10}$$

where $M$ is the number of masked patches, and the ground truth and predicted patches are denoted as $x_i^{(m)}$ and $\hat{x}_i^{(m)}$, respectively.

To complement this, the Cosine Similarity loss ensures that the reconstructed image preserves the high-level semantics of the original. Specifically, both the original $x_i$ and reconstructed $\hat{x}_i$ images are passed through the frozen momentum image encoder $f'_\theta$, and their resulting $[CLS]$ tokens are aligned via cosine similarity:

$$\mathcal{L}_{cos} = 1 - \cos\left(CLS(f_{\theta'}(x_i)), CLS(f_{\theta'}(\hat{x}_i))\right). \tag{11}$$

The Reconstruction Loss combines both components and is defined as:

$$\mathcal{L}_{rec} = \mathcal{L}_{mse} + \mathcal{L}_{cos}. \tag{12}$$

This dual loss design encourages not only pixel-level correctness but also the preservation of identity-specific semantics, which is critical for robust cross-domain generalization.

## 4 EXPERIMENTS

### 4.1 DATASETS AND EVALUATION METRICS

**Multi-camera Datasets.** We evaluate the proposed ReText method on well-known multi-camera datasets: CUHK03-NP (Li et al., 2014), Market-1501 (Zheng et al., 2015), MSMT17 (Wei et al., 2018) with its modification MSMT17-merged, which combines training and testing parts, CUHK-SYSU[1] (Xiao et al., 2017), and a subset of RandPerson (Wang et al., 2020) (Tab. 1). We do not include the multi-camera dataset DukeMTMC-reID (Ristani et al., 2016) in our experiments, as it was withdrawn due to ethical concerns.

Table 1: Statistics of datasets used in our experiments. The single-camera SYNTH-PEDES is orders of magnitude larger than all multi-camera datasets and textually annotated.

| Dataset | Multi | Text | #Images | #IDs | #Cams |
|---|---|---|---|---|---|
| CUHK03-NP | ✓ | ✗ | 14,097 | 1,467 | 2 |
| Market-1501 | ✓ | ✗ | 32,217 | 1,501 | 6 |
| MSMT17 | ✓ | ✗ | 126,441 | 4,101 | 15 |
| CUHK-SYSU | ✓ | ✗ | 34,574 | 11,934 | — |
| RandPerson | ✓ | ✗ | 132,145 | 8,000 | 19 |
| SYNTH-PEDES | ✗ | ✓ | >4.7M | 300K | — |

**Single-camera Dataset.** We use SYNTH-PEDES (Zuo et al., 2024) as the single-camera dataset with textual annotations. It is constructed from images sourced from LUPerson-NL (Fu et al., 2022) and LPW (Song et al., 2018), and enriched with automatically generated captions using a dedicated annotation framework. As seen in Tab. 1, SYNTH-PEDES is significantly larger than traditional multi-camera Re-ID datasets and offers much broader diversity. It is important to note that this single-camera dataset does not overlap with any of the multi-camera datasets used in our experiments, which prevents data leakage and ensures a fair evaluation.

**Metrics.** In our experiments, we use Cumulative Matching Characteristics ($Rank_1$) and mean Average Precision ($mAP$) as evaluation metrics.

### 4.2 IMPLEMENTATION DETAILS

Both image encoder and momentum image encoder are ViT-Base (Dosovitskiy, 2020), pre-trained in a self-supervised manner on the LUPerson dataset (Fu et al., 2021). The text encoder is BERT-Base (Devlin et al., 2019). We use the AdamW optimizer with a learning rate of $10^{-5}$ and a weight decay rate of $0.02$. A warm-up scheme is applied during the first 10 epochs. ReText is trained for 100 epochs with a mini-batch size of 96, comprising 32 images from multi-camera data and 64 from the single-camera data. All images are resized to $256 \times 128$ and augmented with random cropping, horizontal flipping, Gaussian blurring, and random grayscale transformations. More implementation details and analysis of all parameters are provided in Sec. A.1 and Sec. A.3.

### 4.3 PROOF-OF-CONCEPT

We validate two key insights behind ReText: (1) while single-camera data is less challenging for person Re-ID due to the lack of cross-view variation, their stylistic diversity can enhance generalization; and (2) combining them with textual captions provides language supervision that facilitates the learning of domain-invariant representations. Our experimental results in Tab. 2 confirm both claims.

**Single-camera Data is Simple.** The second row of Tab. 2 shows that training solely on single-camera data, without any textual supervision,

Table 2: Effect of single-camera data and textual supervision on cross-domain Re-ID. MSMT17 is used as multi-camera data for training.

| Multi | Single w/o text | Single w. text | CUHK03-NP | | Market-1501 | |
|---|---|---|---|---|---|---|
| | | | $Rank_1$ | $mAP$ | $Rank_1$ | $mAP$ |
| ✓ | | | 48.9 | 49.6 | 91.1 | 77.7 |
| | ✓ | | 24.8 | 26.9 | 84.1 | 66.3 |
| | | ✓ | 36.7 | 36.8 | 88.4 | 71.5 |
| ✓ | ✓ | | 49.9 | 51.3 | 92.4 | 78.2 |
| ✓ | | ✓ | **63.4** | **63.1** | **93.6** | **83.6** |

leads to significantly lower performance compared to standard training on multi-camera Re-ID data (first row). This confirms our assumption that single-camera data, lacking cross-view variation, is inherently simpler and less informative for the Re-ID task.

---

[1] Although each identity in CUHK-SYSU appears under a single camera view, standard evaluation protocols still consider it as multi-camera data.

**Captions Add Complexity.** When textual captions are added to the same single-camera data (third row), cross-domain performance improves significantly. This demonstrates that natural language provides rich semantic cues (e.g., clothing, attributes, actions) that compensate for the lack of cross-view diversity, helping the model learn more discriminative and domain-invariant representations.

**Stylistic Diversity Improves Generalization.**[2] The fourth row of Tab. 2 shows that joint training on a mixture of multi-camera and single-camera data (without text) leads to a clear improvement over using multi-camera data alone (first row). This indicates that the stylistic diversity present in single-camera data exposes the model to a broader distribution of visual appearance, thereby enhancing its ability to generalize to unseen domains.

**Captions And Stylistic Diversity: The Key to Generalization.** Combining stylistic diversity from single-camera data and semantic supervision from captions with multi-camera data (fifth row) leads to the strongest cross-domain performance. This confirms the core idea of ReText: training on both multi-camera and captioned single-camera data enables learning discriminative, semantically grounded, and domain-invariant representations that enhance generalization in person Re-ID.

**Conclusions.** In ReText, the performance gains are driven not simply by adding more image data, but by incorporating textual descriptions and enabling cross-modal learning. Our primary goal is to explore how text supervision during training can enhance image-based person Re-ID, and how semantically rich captions can increase the complexity and usefulness of otherwise simple single-camera data.

## 4.4 Ablation Study

### 4.4.1 Effect of Training Objectives

We first evaluate the impact of each training task by incrementally adding the proposed objectives. As shown in Tab. 3, the baseline model trained only with the Re-ID loss (first row) achieves moderate performance. When we introduce the image-text matching (ITM) task, performance improves substantially across both $Rank_1$ and $mAP$, demonstrating the benefit of leveraging textual supervision from captions. Adding the image reconstruction (IR) task provides further gains. The resulting improvement confirms the robustness of the learned representations, especially under partial visual information and domain shift. Examples of reconstructed images are provided in Sec. A.4. Overall, each proposed objective contributes to cross-domain generalization, with the combination yielding the best results.

Table 3: Step-by-step ablation study of the proposed training objectives. ITM denotes the image-text matching task, and IR denotes the image reconstruction task. The model is trained on MSMT17 and tested on CUHK03-NP.

| Re-ID | ITM | IR | $Rank_1$ | $mAP$ |
|-------|-----|-----|----------|-------|
| ✓ | | | 48.9 | 49.6 |
| ✓ | ✓ | | 62.9 | 62.7 |
| ✓ | ✓ | ✓ | **63.4** | **63.1** |

### 4.4.2 Effect of Matching Loss Design

Since ReText relies heavily on image-text alignment, it is important to examine how the matching objective is formulated. We study the impact of different loss functions for image-text matching in Tab. 4. Replacing the standard CLIP contrastive loss with its soft version (Soft CLIP), which accounts for multiple positive image-text pairs, improves performance. This confirms that soft alignment is beneficial when several captions share the same identity within a mini-batch.

Table 4: Effect of different image-text matching losses on cross-domain Re-ID performance. The model is trained on MSMT17 and tested on CUHK03-NP.

| Loss | $Rank_1$ | $mAP$ |
|------|----------|-------|
| CLIP loss | 59.8 | 60.7 |
| Soft CLIP loss | 60.2 | 61.1 |
| $\mathcal{L}_{im}$ (ours) | 62.2 | 62.3 |
| $\mathcal{L}_{im} + \mathcal{L}_{sp}$ (ours) | **62.9** | **62.7** |

However, our proposed Identity-aware Matching Loss ($\mathcal{L}_{im}$) achieves even better results. Unlike Soft CLIP, which still relies on pairwise similarity,

---

[2]The fourth row of Tab. 2 corresponds to our implementation of ReMix using the same more powerful image encoders as in ReText, as well as the same single-camera data used for training ReText. This setup was chosen to provide a fair and controlled comparison, isolating the effect of textual supervision.

Table 5: Comparison of ReText with other state-of-the-art methods according to Protocol 1. In this table, gray cells indicate experiments that are not applicable.

| Method | Reference | Training Dataset | CUHK03-NP | | Market-1501 | | MSMT17 | |
| --- | --- | --- | --- | --- | --- | --- | --- | --- |
| | | | $Rank_1$ | $mAP$ | $Rank_1$ | $mAP$ | $Rank_1$ | $mAP$ |
| TransMatcher (Liao & Shao, 2021) | NeurIPS | Market-1501 | 22.2 | 21.4 | — | — | 47.3 | 18.4 |
| QAConv-GS (Liao & Shao, 2022) | CVPR | | 19.1 | 18.1 | 91.6 | 75.5 | 45.9 | 17.2 |
| PAT (Ni et al., 2023) | ICCV | | 25.4 | 26.0 | 92.4 | 81.5 | 42.8 | 18.2 |
| LDU (Peng et al., 2024b) | TIM | | 18.5 | 18.2 | — | — | 35.7 | 13.5 |
| DCAC (Li & Gong, 2025) | Sensors | | 33.2 | 32.5 | 94.9 | 86.8 | 52.1 | 23.4 |
| ReMix (Mamedov et al., 2025) | WACV | | — | — | 96.2 | 89.8 | — | — |
| ReText | Ours | | **57.6** | **57.5** | **97.4** | **93.8** | **70.1** | **43.1** |
| TransMatcher (Liao & Shao, 2021) | NeurIPS | MSMT17 | 23.7 | 22.5 | 80.1 | 52.0 | — | — |
| QAConv-GS (Liao & Shao, 2022) | CVPR | | 20.9 | 20.6 | 79.1 | 49.5 | 79.2 | 50.9 |
| PAT (Ni et al., 2023) | ICCV | | 24.2 | 25.1 | 72.2 | 47.3 | 75.9 | 52.0 |
| LDU (Peng et al., 2024b) | TIM | | 21.3 | 21.3 | 74.6 | 44.8 | — | — |
| DCAC (Li & Gong, 2025) | Sensors | | 34.4 | 34.1 | 77.9 | 52.1 | 88.3 | 70.1 |
| ReMix (Mamedov et al., 2025) | WACV | | 27.3 | 27.4 | 78.2 | 52.4 | 84.8 | 63.9 |
| ReText | Ours | | **63.4** | **63.1** | **93.6** | **83.6** | **91.4** | **78.7** |
| QAConv (Liao & Shao, 2020) | ECCV | MSMT17-merged | 25.3 | 22.6 | 72.6 | 43.1 | | |
| TransMatcher (Liao & Shao, 2021) | NeurIPS | | 31.9 | 30.7 | 82.6 | 58.4 | | |
| QAConv-GS (Liao & Shao, 2022) | CVPR | | 27.6 | 28.0 | 82.4 | 56.9 | | |
| PAT (Ni et al., 2023) | ICCV | | 27.4 | 28.7 | 72.8 | 48.6 | | |
| CLIP-DFGS (Zhao et al., 2024) | TOMM | | 37.1 | 25.7 | 80.9 | 55.2 | | |
| ReMix (Mamedov et al., 2025) | WACV | | 37.7 | 37.2 | 84.0 | 61.0 | | |
| ReText | Ours | | **66.1** | **66.8** | **93.9** | **84.7** | | |
| TransMatcher (Liao & Shao, 2021) | NeurIPS | RandPerson | 17.1 | 16.0 | 77.3 | 49.1 | 48.3 | 17.7 |
| QAConv-GS (Liao & Shao, 2022) | CVPR | | 18.4 | 16.1 | 76.7 | 46.7 | 45.1 | 15.5 |
| PAT (Ni et al., 2023) | ICCV | | 20.2 | 20.1 | 73.7 | 46.9 | 45.5 | 19.4 |
| ReMix (Mamedov et al., 2025) | WACV | | 19.3 | 18.4 | 72.7 | 45.4 | — | — |
| ReText | Ours | | **46.9** | **47.7** | **91.7** | **79.7** | **70.8** | **42.3** |

$\mathcal{L}_{im}$ learns the full distribution over all positive and negative instances. This enables richer and more flexible supervision. Finally, adding the Structure-preserving Loss ($\mathcal{L}_{sp}$) further improves the results by enforcing intra-class consistency between image embeddings in the shared embedding space.

## 4.5 COMPARISON WITH STATE-OF-THE-ART METHODS

We compare ReText with other state-of-the-art generalizable person Re-ID methods using three standard evaluation protocols. Protocol 1 trains on a single multi-camera dataset and tests on a different one. Protocol 2 uses several multi-camera datasets for training and evaluates on a hold-out target dataset. Protocol 3 is similar to Protocol 2 but includes both train and test splits of the source datasets for training.

**Protocol 1.** As shown in Tab. 5, ReText consistently outperforms existing methods across all cross-domain benchmarks. A key observation is that the performance gap between training on MSMT17 and its extension MSMT17-merged is relatively small for ReText, while it is much larger for methods like ReMix. This indicates that ReText relies less on the scale of multi-camera data and benefits more from the semantic richness of textual supervision and the stylistic diversity of single-camera data. ReText also surpasses transformer-based models such as TransMatcher (Liao & Shao, 2021) and PAT (Ni et al., 2023), confirming that its generalization strength comes from multimodal joint learning on a mixture of multi-camera and single-camera data rather than architectural complexity. Additionally, it outperforms QAConv (Liao & Shao, 2020), QAConv-GS (Liao & Shao, 2022), and TransMatcher even though those models use larger input image sizes.

Table 6: Comparison of ReText with other state-of-the-art methods according to Protocols 2 and 3. In this table, C3 is CUHK03-NP, MS is MSMT17, M is Market-1501, and CS is CUHK-SYSU. *ReMix is excluded from this comparison because its authors did not evaluate the method using these protocols.*

| | Method | Reference | M+MS+CS → C3 | | M+CS+C3 → MS | | MS+CS+C3 → M | | Average | |
|---|---|---|---|---|---|---|---|---|---|---|
| | | | $Rank_1$ | $mAP$ | $Rank_1$ | $mAP$ | $Rank_1$ | $mAP$ | $Rank_1$ | $mAP$ |
| Protocol 2 | META (Xu et al., 2022) | ECCV | 35.1 | 36.3 | 49.9 | 22.5 | 86.1 | 67.5 | 57.0 | 42.1 |
| | ACL (Zhang et al., 2022) | ECCV | 41.8 | 41.2 | 45.9 | 20.4 | 89.3 | 74.3 | 59.0 | 45.3 |
| | CLIP-ReID (Li et al., 2023) | AAAI | 41.9 | 42.1 | 53.1 | 26.6 | 84.4 | 68.8 | 59.8 | 45.8 |
| | ReFID (Peng et al., 2024a) | TOMM | 34.8 | 33.3 | 39.8 | 18.3 | 85.3 | 67.6 | 53.3 | 39.7 |
| | GMN (Qi et al., 2024) | TCSVT | 42.1 | 43.2 | 50.9 | 24.4 | 87.1 | 72.3 | 60.0 | 46.6 |
| | CLIP-DFGS (Zhao et al., 2024) | TOMM | 51.1 | 50.4 | 59.7 | 31.5 | 90.5 | 79.0 | 67.1 | 53.6 |
| | BAU (Cho et al., 2024) | NeurIPS | 43.9 | 42.8 | 50.9 | 24.3 | 90.4 | 77.1 | 61.7 | 48.1 |
| | CLIP-FGDI (Zhao et al., 2025) | TIFS | 44.6 | 44.4 | 59.4 | 31.1 | 91.3 | 79.4 | 65.1 | 51.6 |
| | ReText | Ours | **65.6** | **65.7** | **63.4** | **39.3** | **91.3** | **82.1** | **73.4** | **62.4** |
| Protocol 3 | META (Xu et al., 2022) | ECCV | 46.2 | 47.1 | 52.1 | 24.4 | 90.5 | 76.5 | 62.9 | 49.3 |
| | ACL (Zhang et al., 2022) | ECCV | 50.1 | 49.4 | 47.3 | 21.7 | 90.6 | 76.8 | 62.7 | 49.3 |
| | CLIP-ReID (Li et al., 2023) | AAAI | 45.8 | 44.9 | 52.6 | 26.8 | 83.4 | 67.5 | 60.6 | 46.4 |
| | ReFID (Peng et al., 2024a) | TOMM | 44.2 | 45.5 | 43.3 | 20.6 | 87.9 | 72.5 | 58.5 | 46.2 |
| | GMN (Qi et al., 2024) | TCSVT | 50.1 | 49.5 | 51.0 | 24.8 | 89.0 | 75.9 | 63.4 | 50.1 |
| | CLIP-DFGS (Zhao et al., 2024) | TOMM | 51.3 | 51.6 | 62.0 | 33.4 | 91.6 | 81.0 | 68.3 | 55.3 |
| | BAU (Cho et al., 2024) | NeurIPS | 51.8 | 50.6 | 54.3 | 26.8 | 91.1 | 79.5 | 65.7 | 52.3 |
| | CLIP-FGDI (Zhao et al., 2025) | TIFS | 50.1 | 50.1 | 61.4 | 32.9 | 91.2 | 79.8 | 67.6 | 54.3 |
| | ReText | Ours | **68.4** | **69.2** | **64.6** | **40.3** | **93.2** | **84.1** | **75.4** | **64.5** |

**Protocols 2 and 3.** When trained on several multi-camera datasets, ReText demonstrates strong and consistent performance across all target domains (Tab. 6), highlighting its ability to benefit not only from stylistic diversity in single-camera data and semantic supervision provided by textual descriptions, but also from domain variation in multi-camera datasets. Notably, ReText significantly outperforms CLIP-ReID (Li et al., 2023), CLIP-DFGS (Zhao et al., 2024), and CLIP-FGDI (Zhao et al., 2025), which rely on learnable text tokens instead of descriptive captions associated with images. This comparison confirms our hypothesis that the semantic richness and generalization capability of natural language remain underutilized in current image-based person Re-ID approaches. By directly leveraging textual descriptions, ReText effectively incorporates semantic cues that substantially boost cross-domain generalization.

**Comparison Conclusions.** Only ReMix and our ReText are trained using additional single-camera data, which is central to the design of both methods. All other approaches rely solely on labeled multi-camera datasets, as they are not designed to handle heterogeneous training data. Therefore, adapting them to use single-camera data would lead to an unfair degradation of their performance. For this reason, their results are reported under the official training settings. Moreover, ReText is the first method that uses textual descriptions for simple single-camera data by design. Our experiments show that ReMix is less effective at leveraging single-camera data compared to ReText, which further highlights the advantage of the proposed method.

## 5 CONCLUSION

In this paper, we proposed ReText, the novel method for generalizable image-based person Re-ID that is trained on a mixture of multi-camera Re-ID data and single-camera data with textual descriptions. Unlike prior work that ignores single-camera data or underuses language, ReText effectively combines both to learn discriminative, domain-invariant representations. It jointly optimizes three tasks: Re-ID, image-text matching, and text-guided image reconstruction. Experiments across three standard generalization protocols show that adding semantic supervision from text and stylistic diversity from single-camera data notably improves cross-domain performance. We believe our work will serve as a basis for future research dedicated to generalized, accurate, and reliable person Re-ID.

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

# A  APPENDIX

## A.1  SOME ADDITIONAL IMPLEMENTATION DETAILS

### A.1.1  MINI-BATCH COMPOSITION

Each mini-batch in ReText consists of a mixture of multi-camera and single-camera data:

- For the multi-camera portion, we use the well-known $PK$ sampling strategy (Hermans et al., 2017), following ReMix (Mamedov et al., 2025): $P_m = 8$ person identities are randomly selected, and for each identity, $K_m = 4$ images from different cameras are sampled, resulting in $N_m = P_m \times K_m = 8 \times 4 = 32$ images.

- For the single-camera portion, we similarly apply the $PK$ sampler with $P_s = 32$ and $K_s = 2$, but without enforcing camera diversity, since each identity appears in only one camera. This yields an additional $N_s = 64$ images per batch.

In total, each mini-batch consists of 96 images: 32 from multi-camera data and 64 from single-camera data. A detailed analysis of the sampling parameters for single-camera data is provided in the following sections (Sec. A.3.1 and Sec. A.3.2).

### A.1.2  DECODER ARCHITECTURE

The decoder, used for the image reconstruction task (Fig. 2), is responsible for predicting missing image patches based on visible image patches and textual tokens. It consists of four repeated transformer-based decoder blocks, each comprising: a self-attention layer; a cross-attention layer, enabling information flow from the text tokens; a feed-forward network. Finally, all output tokens are projected into pixel space via a linear layer, which reconstructs the masked patches of the image.

## A.2  USING SINGLE-CAMERA DATA IN RE-ID LOSSES

In ReText, we adopt the Instance ($\mathcal{L}_{ins}$), Augmentation ($\mathcal{L}_{aug}$), Centroids ($\mathcal{L}_{cen}$), and Camera Centroids ($\mathcal{L}_{cc}$) losses from ReMix for the Re-ID task. Unlike ReMix, which applies these loss functions (except for $\mathcal{L}_{cc}$) to both multi-camera and single-camera data, ReText restricts their use to multi-camera data only. As justified in the main paper, Re-ID supervision is most effective when cross-view variation is present, which is inherently lacking in single-camera data. The latter is stylistically diverse but considerably simpler and less informative for direct cross-view identity classification.

Table 7: Impact of using single-camera data in different Re-ID loss functions. The model is trained on MSMT17.

| Configuration | Text | CUHK03-NP | | Market-1501 | |
|---|---|---|---|---|---|
| | | $Rank_1$ | $mAP$ | $Rank_1$ | $mAP$ |
| in all Re-ID losses | ✗ | 49.9 | 51.3 | 92.4 | 78.2 |
| in no Re-ID losses | ✓ | 63.4 | **63.1** | **93.6** | **83.6** |
| only in $\mathcal{L}_{ins}$ | ✓ | 63.1 | 62.9 | **93.6** | **83.6** |
| only in $\mathcal{L}_{aug}$ | ✓ | 62.6 | 62.3 | 93.5 | 83.3 |
| only in $\mathcal{L}_{cen}$ | ✓ | **64.2** | 62.8 | 93.3 | 82.3 |

To verify this hypothesis, we conduct a series of experiments in which Re-ID loss functions are selectively applied to single-camera data. The results are summarized in Tab. 7. In most cases, adding Re-ID supervision to single-camera data leads to a degradation in cross-domain performance, confirming our design choice.

The first two rows of the table are especially instructive. In the first case, the model is trained without text supervision, and Re-ID losses are applied to both multi-camera and single-camera data. In the second case, we introduce textual supervision but apply Re-ID losses only to multi-camera data. The significant performance gain observed in the second setting highlights a key insight: while single-camera data is less complex for person Re-ID, pairing it with descriptive captions unlocks rich semantic cues that significantly enhance domain generalization. The best results are achieved when single-camera data is excluded from Re-ID losses and instead used only for auxiliary tasks such as image-text matching and reconstruction.

### A.3 PARAMETER ANALYSIS

### A.3.1 EFFECT OF SAME-IDENTITY INSTANCES IN A MINI-BATCH

To determine the optimal number of same-identity instances from single-camera data in a mini-batch, we analyze different image-text matching loss functions under varying values of $K_s$ (Tab. 8). As shown in the results, performance under the standard CLIP contrastive loss decreases as $K_s$ increases. This is expected, as CLIP performs strict one-to-one matching and cannot effectively handle multiple positive image-text pairs per identity in a mini-batch. In contrast, Soft CLIP loss — which accounts for multiple positive image-text pairs — shows improved performance when $K_s = 2$, confirming that soft alignment is beneficial in this setting. The best performance is achieved using our Identity-aware Matching Loss ($\mathcal{L}_{im}$) with $K_s = 2$. This highlights not only its robustness for image-text alignment but also its flexibility in handling multiple positive image-text pairs per identity in a mini-batch.

Table 8: Analysis of the number of same-identity instances ($K_s$) from single-camera data in a mini-batch for different image-text matching loss functions. The model is trained on MSMT17 and tested on CUHK03-NP.

| Loss | $K_s$ | $Rank_1$ | $mAP$ |
|---|---|---|---|
| CLIP loss | 1 | 59.8 | 60.3 |
|  | 2 | 59.6 | 60.1 |
|  | 4 | 59.5 | 59.2 |
| Soft CLIP loss | 1 | 59.8 | 60.3 |
|  | 2 | 60.0 | 60.5 |
|  | 4 | 59.0 | 59.5 |
| $\mathcal{L}_{im}$ (ours) | 1 | 60.6 | 60.6 |
|  | 2 | **61.4** | **61.3** |
|  | 4 | 60.4 | 60.0 |

**Note:** in these experiments, the number of single-camera images in a mini-batch ($N_s$) was fixed to 32, and we used $\alpha = 0.5$ in Eq. 6. Therefore, selecting $K_s = 2$ implies that $N_{pos} = 1$ in Eq. 6.

### A.3.2 NUMBER OF SINGLE-CAMERA IMAGES IN A MINI-BATCH

We study how the number of single-camera images in a mini-batch ($N_s$) affects cross-domain Re-ID performance. As shown in Tab. 9, increasing $N_s$ from 32 to 64 and 128 leads to gradual improvements. This trend suggests that more single-camera data — when paired with descriptive captions — provides stronger semantic supervision and greater visual diversity, thereby improving generalization. However, the gains saturate beyond 64 images, indicating a trade-off between diversity and computational efficiency.

Table 9: Analysis of the number of single-camera images in a mini-batch ($N_s$). The model is trained on MSMT17 and tested on CUHK03-NP.

| $N_s$ | $Rank_1$ | $mAP$ |
|---|---|---|
| 32 | 61.4 | 61.3 |
| 64 | 61.6 | **61.9** |
| 128 | **61.9** | 61.8 |

**Note:** in these experiments, the number of same-identity instances ($K_s$) from single-camera data in a mini-batch was fixed to 2, and we used $\alpha = 0.5$ in Eq. 6.

### A.3.3 CONTROLLING POSITIVE IMAGE-TEXT PAIRS WEIGHTING WITH $\alpha$

We analyze the effect of the $\alpha$ parameter in Eq. 6, which controls the relative weight of the main image-text pair versus other positive pairs in the Identity-aware Matching Loss. As shown in Tab. 10, the best performance is achieved with $\alpha = 0.6$, indicating that slightly prioritizing the anchor pair while still leveraging additional positives yields the most robust alignment.

Overweighting additional captions (i.e., setting lower $\alpha$) can be risky: although the images refer

Table 10: Analysis of the $\alpha$ parameter in Eq. 6 for the Identity-aware Matching Loss. The model is trained on MSMT17 and tested on CUHK03-NP.

| $\alpha$ | $Rank_1$ | $mAP$ |
|---|---|---|
| 0.5 | 61.6 | 61.9 |
| 0.6 | **62.2** | **62.3** |
| 0.7 | 61.9 | 62.0 |

to the same identity, the corresponding textual descriptions may differ due to variations in pose, visibility, or context. For instance, sneakers might be visible in one image but occluded in another, resulting in different caption content. Nevertheless, these additional captions provide valuable com-

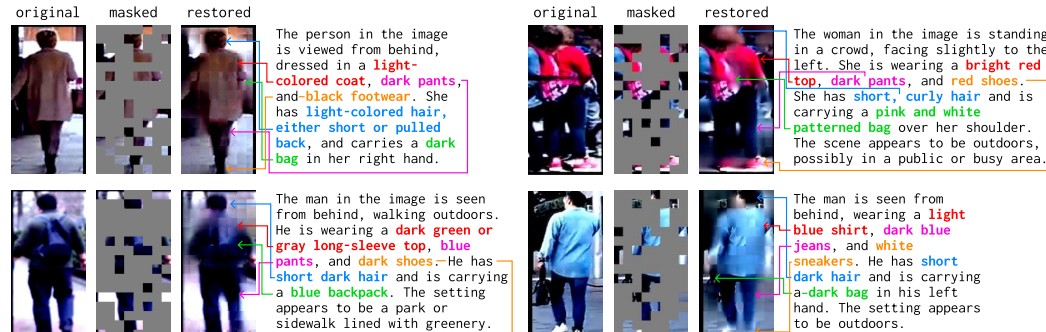

Figure 3: Examples of image reconstruction in ReText. Each example shows the original image (left), masked input (middle), and the reconstructed output (right), alongside the corresponding caption. The model successfully recovers key visual details, highlighting the effectiveness of image-text training.

plementary semantic cues. Our loss formulation strikes a balance — leveraging semantic diversity across views while preserving stability in the matching process.

**Note:** in these experiments, the number of single-camera images in a mini-batch ($N_s$) was fixed to 64, and we used $K_s = 2$.

### A.3.4 EFFECT OF THE TEMPERATURE PARAMETER $\tau$ IN THE STRUCTURE-PRESERVING LOSS

We study the influence of the temperature parameter $\tau$ in Eq. 9, which controls the sharpness of similarity distributions in the Structure-preserving Loss. As shown in Tab. 11, setting $\tau = 0.1$ yields the best performance. This value provides a good trade-off between emphasizing hard positives and maintaining stable gradients.

Too small a value (e.g., $\tau = 0.07$) makes the loss overly sensitive to small similarity differences, while larger values (e.g., $\tau \geq 0.15$) overly smooth the similarity scores, weakening the structure-preserving effect. These results confirm that fine-tuning $\tau$ is crucial for enforcing intra-class structure without destabilizing training.

Table 11: Analysis of the temperature parameter $\tau$ in Eq. 9 for the Structure-preserving Loss. The model is trained on MSMT17 and tested on CUHK03-NP.

| $\tau$ | $Rank_1$ | $mAP$ |
|---|---|---|
| 0.07 | 62.6 | 62.5 |
| 0.1 | **62.9** | **62.7** |
| 0.15 | 61.9 | 62.3 |
| 0.2 | 61.9 | 61.8 |

**Note:** in these experiments, the number of single-camera images in a mini-batch ($N_s$) was fixed to 64, $K_s = 2$, and we used $\alpha = 0.6$.

### A.4 IMAGE RECONSTRUCTION

In our experiments, we mask $75\%$ of image patches. As illustrated in Fig. 3, ReText successfully reconstructs the human silhouette and key attributes. These examples also highlight the influence of text supervision: the model is able to recover attributes that are entirely masked out in the image, guided by their presence in the accompanying description. It is important to note that reconstruction serves as an auxiliary task — our goal is not to achieve photorealistic quality but to improve generalization. As shown in Tab. 3, the image reconstruction task enhances cross-domain Re-ID performance by helping ReText learn robust representations under partial visual information and domain shift.

## A.5 FUTURE RESEARCH

This work pioneers the use of textual descriptions in image-based person Re-ID during training, demonstrating that semantically rich captions can significantly improve cross-domain generalization. Although achieving photorealistic quality was not our primary objective, experiments show that ReText can reliably recover human silhouettes and key attributes when guided by informative textual descriptions. Future research could explore improving the granularity and consistency of textual descriptions to further enhance both Re-ID accuracy and reconstruction quality. It could also investigate noise reduction in automatically generated descriptions and assess its impact on quality.

Moreover, our findings indicate that single-camera data — when paired with descriptive text — becomes a valuable source of diversity and supervision. However, optimal sampling strategies from such data remain an open question. Exploring better mini-batch composition, instance balancing, or caption-aware sampling may lead to further gains.

Finally, while this study focuses on using textual supervision primarily for single-camera data, an exciting direction is to investigate the role of captions for multi-camera Re-ID data, where textual information could complement cross-view variation and support fine-grained alignment.

