# OpenReview forum: "ReText: Text Boosts Generalization in Image-Based Person Re-identification"
_ICLR.cc/2026/Conference — Submitted to ICLR 2026_

### Official Review · Reviewer_EKey · 2025-10-27

**Soundness:** 3
**Presentation:** 3
**Contribution:** 2
**Rating:** 4
**Confidence:** 4

**Summary:**

The paper presents ReText, a method for generalizable person re-identification (Re-ID) to recognize individuals across cameras in unseen domains without retraining. It combines multi-camera Re-ID data with single-camera data enhanced by textual descriptions. ReText trains on three tasks: Re-ID, image-text matching, and text-guided image reconstruction. Experiments show it outperforms limited existing methods on cross-domain Re-ID benchmarks.

**Strengths:**

1. ReText introduces a pioneering framework by integrating multi-camera Re-ID data with single-camera data enriched by textual descriptions. The joint optimization of three tasks, Re-ID, image-text matching, and text-guided image reconstruction, is a fresh perspective in the Re-ID domain, leveraging multimodal learning to enhance generalization.
2. By complementing single-camera data with textual descriptions, ReText addresses the limitation of minimal cross-view variation in such data, potentially enriching semantic cues and improving robustness across unseen domains.
3. The method demonstrates significant improvements over state-of-the-art (SOTA) methods on cross-domain Re-ID benchmarks, highlighting its potential to handle domain gaps effectively.
4. Public Code Availability: The commitment to releasing code publicly enhances reproducibility and encourages further research in this direction.

**Weaknesses:**

1. Missing Common Evaluation Settings: While ReText compares against some SOTA methods, it lacks results for standard settings, such as fully supervised performance after downstream fine-tuning on target domains. This omission makes it difficult to fully assess the method’s practical utility and robustness compared to established Re-ID approaches.

2. Limited Novelty Perception: The proposed method bears similarities to existing frameworks like Masked Autoencoders (MAE) and CLIP, which may reduce its perceived innovation. The authors need to better articulate the unique aspects of their joint optimization strategy to differentiate it from these prior works.

3. Lack of Generated Data Comparisons: The study does not compare ReText’s approach with other generated data methods, such as MALS or LUPerson-NL only, limiting the ability to evaluate the effectiveness of its textual augmentation strategy relative to existing data augmentation techniques.

4. Insufficient Clarity on Experimental Settings: Key experimental details, such as specific training configurations or evaluation protocols, are relegated to the supplementary material. This placement risks misinterpretation of the method’s performance and reduces the main text’s clarity and impact.

**Questions:**

1. Downstream Fine-Tuning Results: Why were fully supervised results after downstream fine-tuning on target domains not included? Could the authors provide these results to better contextualize ReText’s performance against SOTA methods in standard Re-ID settings?

2. Novelty Differentiation: How does ReText’s joint optimization of Re-ID, image-text matching, and text-guided reconstruction differ fundamentally from combining MAE and CLIP-like frameworks? Can the authors clarify the specific innovations in their approach?

3. Comparison with Generated Data: Why were comparisons with prior generated data approaches (e.g., MALS, LUPerson-caption only) not included? How does ReText’s textual description strategy compare to these methods in terms of semantic enrichment and generalization?

4. Placement of Experimental Details: Why were critical experimental settings (protocol) placed in the supplementary material rather than the main text? Would moving these details to the main paper improve clarity and prevent potential misinterpretation of the results?

5. Task Interaction Mechanisms: How are the three tasks balanced during joint optimization? Are there specific weighting strategies or loss functions that ensure effective learning, and how do they impact the overall generalization performance?

6. The proposed Structure-preserving Loss is still a InfoNCE loss in the image modality;  the identity matching loss is a soft cross-entropy loss, which is quite common for smooth labels.

7. Missing some general person retrieval methods like All in one framework for multimodal re-identification in the wild (CVPR), Adaptive Uncertainty-Based Learning for Text-Based Person Retrieval (AAAI)

---

> ### Author Response · Authors · 2025-11-19
> **Weakness 1 and Question 1**
>
> _No comparison with SOTA in the standard person Re-ID task (weakness 1 and question 1)._
>
> We acknowledge the reviewer’s concern regarding standard evaluation settings. Although our main focus is on the **generalization** scenario of image-based person Re-ID — where training and testing are conducted on different datasets — we also include experiments under the standard single-domain evaluation protocol.
>
> As shown in Table 5, we report downstream fine-tuning performance on both Market-1501 and MSMT17 for all comparable methods (with the exception of TransMatcher and LDU, which do not provide results under these settings). Please refer to the corresponding row-column intersections for Market-1501 and MSMT17, respectively.

---

> ### Author Response · Authors · 2025-11-19
> **Weakness 2 and Question 2**
>
> In our method, we do not use MAE or CLIP techniques _as-is_. For example, in ReText, the image reconstruction task incorporates not only unmasked visual patches (as in MAE) but also **textual descriptions**, which provide an additional source of semantic guidance during reconstruction.
>
> Regarding the image–text matching task, **we introduce two new loss functions** — Identity-aware Matching Loss and Structure-preserving Loss — **which explicitly account for the specifics of person re-identification and the composition of mini-batches during training.** In our experiments, we show that these proposed loss functions lead to better Re-ID performance than simply using the standard CLIP loss (Section 4.4.2, Effect of Matching Loss Design).

---

> ### Author Response · Authors · 2025-11-19
> **Weakness 3 and Question 3**
>
> _About using MALS and LUPerson-NL (weakness 3 and question 3)._
>
> We thank the reviewer for the questions. We do not compare our method with LUPerson or LUPerson-NL because these **datasets do not contain textual descriptions**, which are a fundamental component of our approach. For this reason, our method uses SYNTH-PEDES, which provides both images and text. In addition, **the authors of LUPerson and LUPerson-NL did not study the effect of training on these datasets on generalization performance, nor did they evaluate their methods in a cross-dataset setting** — while cross-dataset generalization is the primary focus of our work.
>
> However, even though ReText is not designed for the standard single-dataset setting (training and testing on the same dataset), we can still make a comparison using the results reported for LUPerson and LUPerson-NL on Market-1501 and MSMT17. From this comparison, we observe:
>
> - The best model pre-trained on **LUPerson** and fine-tuned on Market-1501 achieves **91.0 mAP**; when fine-tuned on MSMT17, it achieves **65.7 mAP**.
>
> - The best model pre-trained on **LUPerson-NL** and fine-tuned on Market-1501 achieves **91.9 mAP**; when fine-tuned on MSMT17, it achieves **68.0 mAP**.
>
> - In contrast, **ReText**, trained jointly on SYNTH-PEDES and Market-1501, achieves **93.8 mAP** on Market-1501; when trained jointly on SYNTH-PEDES and MSMT17, it achieves **83.6 mAP** on MSMT17.
>
> Moreover, the following two points are important to highlight:
>
> 1. **SYNTH-PEDES is comparable in scale to LUPerson**, containing roughly 4 million person images.
>
> 2. **SYNTH-PEDES is a subset of LUPerson-NL.** Therefore, the improvements provided by ReText do not stem from larger data volume, but from the **proposed training strategy**.
>
> Additionally, ReText uses a backbone **pre-trained on LUPerson**. If we examine Table 2 in Section 4.3 (Proof-of-Concept) and compare the first and fifth rows, we see that the proposed strategy of **joint training on multi-camera and text-enriched single-camera data** yields significantly better performance than standard self-supervised pre-training — further confirming the effectiveness of our approach.
>
> Regarding **MALS, we do not include a comparison because that work addresses text-based person Re-ID**, whereas our focus is on improving **generalization in image-based person Re-ID**. These are fundamentally different tasks that are not directly comparable.

---

> ### Author Response · Authors · 2025-11-19
> **Weakness 4 and Question 4**
>
> _Why were critical experimental settings (protocol) placed in the supplementary material rather than the main text? Would moving these details to the main paper improve clarity and prevent potential misinterpretation of the results? (weakness 4 and question 4)_
>
> We fully acknowledge the importance of presenting complete experimental configurations directly in the main paper. However, due to the strict 9-page limit, we had to prioritize essential content such as the proof-of-concept analysis, comprehensive ablation studies, and comparisons with existing methods.
>
> **During the discussion phase and in the camera-ready version, the page limit is increased to 10 pages according to the conference rules. Therefore, we have returned Table 6, which contains the configurations for Protocols 2 and 3, into the main paper.**

---

> ### Author Response · Authors · 2025-11-19
> **Question 5**
>
> _Task Interaction Mechanisms: How are the three tasks balanced during joint optimization? Are there specific weighting strategies or loss functions that ensure effective learning, and how do they impact the overall generalization performance? (question 5)_
>
> We thank the reviewer for the thoughtful questions. In ReText, **all three tasks contribute equally to the final objective**, and we agree that this point should be explicitly highlighted in the camera-ready version.
>
> Regarding the loss functions, our method is trained using the Re-ID losses **carefully designed in ReMix**, along with **two newly proposed loss functions for image-text matching**: Identity-aware Matching Loss and Structure-preserving Loss. **Our experiments demonstrate that these losses are more effective than the standard CLIP loss** (see Table 4). This empirical evidence shows that the proposed losses are successfully adapted to the specifics of the person re-identification problem.
>
> For text-guided image reconstruction, **we also introduce a dedicated Reconstruction Loss**, which combines pixel-level MSE with cosine similarity in the embedding space (Section 3.4.2, The Reconstruction Loss).
>
> As for the influence of the proposed tasks and loss functions on generalization, Table 3 shows that **all components introduced in ReText substantially improve performance in the cross-dataset scenario**. This clearly confirms an increase in generalization capability, since in a cross-dataset setting the training and testing domains are different.

---

> ### Author Response · Authors · 2025-11-19
> **Question 6**
>
> _The proposed Structure-preserving Loss is still a InfoNCE loss in the image modality; the identity matching loss is a soft cross-entropy loss, which is quite common for smooth labels (question 6)._
>
> We thank the reviewer for the insightful remarks. We would like to offer additional clarification. While the proposed Structure-preserving Loss may appear similar to InfoNCE at first glance, **it is fundamentally different**: our loss pulls each anchor toward its **hardest positive**, thereby making the objective more challenging and encouraging the model to learn more discriminative features.
>
> Regarding the Identity-aware Matching Loss, **it is not equivalent to a soft cross-entropy loss.** Instead, it is formulated as a **KL divergence** between the predicted distribution and a specially constructed **soft target distribution**. This target distribution is designed to account for the contribution of both the main positive image-text pair and other positive pairs within the batch.
>
> This design addresses a key challenge: in a training mini-batch, multiple images and textual descriptions may correspond to the same identity. The standard CLIP loss only pulls together the ground-truth (main) image–text pair and treats all remaining pairs as negatives — even when several of those “negatives” actually describe the same person. This is a strong and often harmful assumption. An alternative would be to treat all image-text pairs of the same identity as positives with equal weights, but this approach is problematic because textual descriptions associated with other images of the same person may not fully correspond to the specific visual appearance of the current image.
>
> To handle this, **Identity-aware Matching Loss introduces a weight α for the main positive pair**, allowing it to dominate the target distribution while still incorporating other valid positives.
>
> It is also important to emphasize that **Identity-aware Matching Loss and Structure-preserving Loss work synergistically and complement one another**: the former **improves the quality of image-text matching**, while the latter **preserves intra-class structure within the shared embedding space** (Table 4).

---

> ### Author Response · Authors · 2025-11-19
> **Question 7**
>
> _Missing some general person retrieval methods like All in one framework for multimodal re-identification in the wild (CVPR), Adaptive Uncertainty-Based Learning for Text-Based Person Retrieval (AAAI) (question 7)._
>
> We will be glad to include these works in the Related Work section of the camera-ready version.

---

> ### Comment · Reviewer_EKey · 2025-11-26
>
> Thank you.
>
> The author does not address my concerns on novelty, especially the technical contribution.
>
> The experiment is not very fair. I also agree LUPerson is not an easy dataset, but a strong pretraining dataset. Some works are already caption the LUPerson with text description.
>
> MALS and other text-based person retrieval datasets are needed to be compared.

---

> > ### Author Response · Authors · 2025-11-26
> >
> > Dear Reviewer,
> >
> > Thank you for your feedback. Since the discussion phase is still ongoing, we would be very grateful if you could provide more details regarding your remaining concerns about the novelty of our work. We believe we can clarify these points and further improve the paper based on your input.
> >
> > Thank you again for your time and engagement.

---

### Official Review · Reviewer_2HAB · 2025-10-31

**Soundness:** 3
**Presentation:** 2
**Contribution:** 1
**Rating:** 0
**Confidence:** 5

**Summary:**

The paper aims to address the problem of generalizable image-based person re-identification. It points out that existing methods have found incorporating single-camera data to be beneficial. However, single-camera data lacks cross-view variation, which is a natural limitation. Logically, the problem to solve is this very deficiency. The paper then states that “single-camera data is less complex due to limited cross-view variation,” implying that increasing complexity is the key. The question then becomes: how to enhance this complexity? The authors propose that “pairing single-camera data with descriptive captions unlocks rich semantic cues.”
Intuitively, this suggests that the paper should focus on pairing the data with high-quality captions to maximize semantic richness. But, interestingly, the paper quickly points out that such datasets already exist—specifically, SYNTH-PEDES. This raises the question: what exactly does this paper contribute beyond the existence of SYNTH-PEDES?
The authors summarize their contribution as simultaneously addressing two major challenges: first, the scarcity of multi-camera data, which they mitigate by leveraging stylistically diverse single-camera data; and second, the underutilization of natural language supervision. By effectively incorporating the pre-existing SYNTH-PEDES dataset, the paper’s core work lies in designing losses and modules to fully exploit this resource.
This approach proves successful, as evidenced by their experimental results showing that ReText achieves very strong performance.

**Strengths:**

[+] GOOD performance.

**Weaknesses:**

[-] The paper offers almost no novelty. The losses and modules employed are off-the-shelf, and the dataset used is already publicly available.
[-] The motivation is somewhat confused. As the summary states, the paper argues that “single-camera data is less complex due to limited cross-view variation.” But why does the lack of cross-view necessarily imply less complexity? Take LUPerson, for example—a dataset collected from numerous videos across diverse scenes. Surely, that is complex data.
I agree that pairing images with captions can help explore additional complexity. However, if captioning data is so beneficial, why is multi-camera data left without captions? It could just as well be captioned. Adding captions to multi-camera data would: (1) increase data distribution complexity and improve generalization, and (2) simplify the model architecture by treating single-camera and multi-camera data uniformly, merging them into one dataset and using a shared pipeline.
Ultimately, the paper’s reliance on single-camera data with captions stems simply from the fact that multi-camera datasets lack ready-made captions. Therefore, the paper does not contribute anything new regarding dataset construction or augmentation.
In summary, the so-called challenges the paper claims to tackle have already been addressed by prior works. This paper mainly attempts to integrate existing components in a coherent manner.
[-] Although the experimental results are strong, the conclusions drawn are neither novel nor deep. The claims that “Single-camera data is simple” and “Captions add complexity” have been observed elsewhere. Similarly, conclusions like “Stylistic diversity improves generalization” are well-established in earlier domain generalization research. This paper does not offer particularly insightful perspectives.
[-] The text captions in the SYNTH-PEDES dataset used in this paper are generated by a captioner trained on CUHK and ICFG datasets. Since the images in CUHK and ICFG originate from CUHK and MSMT—datasets that overlap with the target domain—there is a significant risk of data leakage in the experiments presented. This overlap raises concerns about the validity and generalizability of the reported results.

**Questions:**

See Weaknesses

**Details Of Ethics Concerns:**

none.

---

> ### Author Response · Authors · 2025-11-19
> **Comment 1**
>
> We thank the reviewer for the detailed feedback. With this response, we aim to clarify the key points that may have caused misunderstanding, and we hope that our explanation helps the reviewer better appreciate the contributions of our work.

---

> ### Author Response · Authors · 2025-11-19
> **Comment 2**
>
> _1. “Single-camera data is a complex data.”_
>
> You argue that datasets like LUPerson are collected from diverse scenes and therefore should not be considered “simple.” **We fully agree that scene diversity in these datasets is high. However, this does _not_ contradict our claim.**
>
> The simplicity we refer to is **specific to the task of image-based person Re-ID**, which fundamentally requires:
>
> - the same person appearing across **different cameras**
>
> - with **significant cross-view variation** (viewpoints, illumination, sensor characteristics, etc.)
>
> **Single-camera datasets — LUPerson, LUPerson-NL, SYNTH-PEDES, and others — do not contain such cross-view variability.** Each identity appears _only under one camera_, typically with very limited viewpoint variation.
>
> Thus, from the perspective of **multi-camera image-based person Re-ID**, these datasets are indeed _simple_, even though they are _diverse_ in terms of scenes. **This is empirically shown in Table 2 (Section 4.3, Proof-of-Concept):** using single-camera data “as is” _degrades_ multi-camera Re-ID performance (row 2). This observation was also explicitly highlighted in ReMix.
>
> Therefore, single-camera datasets are:
>
> - **diverse in scenes**, but
>
> - **simple for multi-camera Re-ID**, because they lack cross-view complexity.
>
> _2. “Why not caption multi-camera data?”_
>
> We agree that captioning multi-camera datasets could potentially further enhance generalization. This is indeed an interesting direction for future research.
>
> However, our work focuses on a **different hypothesis**, building upon ReMix:
>
> - ReMix showed that mixing multi-camera and single-camera data improves generalization.
>
> - But single-camera data remain **too simple** (limited cross-view variation).
>
> - Therefore, we propose the first approach in image-based Re-ID that uses **multi-camera data + text-enriched single-camera data**.
>
> **Our goal is to demonstrate that text can increase the complexity of single-camera data, addressing a key limitation of ReMix.** Section 4.3 (Proof-of-Concept) confirms that textual descriptions significantly improve generalization.
>
> Captioning multi-camera datasets would require new annotated datasets and falls outside the scope of our work — but we agree it is a promising future direction that our work may motivate.
>
> _3. “There is no novelty.”_
>
> We respectfully disagree. To the best of our knowledge:
>
> 1. **The only prior work studying joint training on multi-camera and single-camera data in image-based person Re-ID is ReMix.**
>
> 2. **No prior work has analyzed the complexity limitations of single-camera data in this context.**
>
> 3. Before ReMix, single-camera datasets (LUPerson, LUPerson-NL, SYNTH-PEDES, etc.) **were used only for self-supervised pre-training, never for fine-tuning.**
>
> 4. **No prior work has proposed adding textual supervision to single-camera data** to address its lack of cross-view variation in the image-based person Re-ID task.
>
> Therefore, the main ideas of ReText:
>
> - analyzing the limitations of single-camera data,
>
> - enriching them with textual descriptions,
>
> - and jointly training multi-camera + text-enriched single-camera data represent **new contributions that have not been explored previously** in image-based person Re-ID.
>
> _4. “The conclusions are neither novel nor deep.”_
>
> We kindly ask the reviewer to share specific references where these conclusions were previously shown _for image-based person Re-ID_. To the best of our knowledge, such works do not exist.
>
> You cite Section 4.3 of our paper as containing “well-known observations,” but that section actually **demonstrates our claims experimentally**, including the relationships:
>
> - **“Single-camera data is simple”** → confirmed by degraded performance in row 2 of Table 2.
>
> - **“Captions add complexity”** → confirmed by improvements from row 4 → row 5.
>
> - **“Stylistic diversity helps generalization”** → also studied within our controlled setup.
>
> Section 4.3 provides a full logical chain:
>
> 1. Single-camera data alone hurt generalization (simple).
>
> 2. Smart mixing them with multi-camera data improves generalization (ReMix effect).
>
> 3. But they remain too simple.
>
> 4. **Adding text increases complexity** → ReText achieves the best generalization.
>
> To the best of our knowledge, this analysis and these conclusions have **not** been shown previously for this task.
>
> _5. Summary_
>
> Our work contributes a **new training strategy for image-based person Re-ID**, which has not been proposed before:
>
> - **joint training on multi-camera data and text-enriched single-camera data**,
>
> - using a **three-task learning paradigm** (Re-ID, image-text matching, text-guided reconstruction),
>
> - together with **new loss functions** tailored to person Re-ID.
>
> We hope that these clarifications address the concerns raised, and we respectfully believe they justify a higher evaluation of our work.

---

> ### Author Response · Authors · 2025-11-19
> **Comment 3**
>
> We would also like to address the reviewer’s concerns regarding a potential **data leak**. We appreciate the reviewer’s careful analysis, but we respectfully disagree with the conclusion. Below we clarify the situation in detail.
>
> 1. **Image-level leakage does not occur.** The SYNTH-PEDES dataset is constructed from the image sets of LUPerson-NL and LPW. **None of these images overlap with the multi-camera datasets used for evaluation** — CUHK03-NP, Market-1501, MSMT17, CUHK-SYSU, or RandPerson. Therefore, at the visual data level, **no leakage is possible**.
>
> 2. **Captioner training data do not introduce leakage into ReText.** It is true that the captioner used to generate textual descriptions for SYNTH-PEDES was trained on CUHK-PEDES and ICFG-PEDES, and these datasets partly overlap with CUHK03 and MSMT17. However — and this is crucial — **the captioner is not used anywhere inside ReText.** ReText only consumes the **final textual descriptions** produced by the captioner for images from LUPerson-NL and LPW.\
>    Thus:
>
>    - ReText does **not** use the captioner model.
>
>    - ReText does **not** access CUHK-PEDES or ICFG-PEDES.
>
>    - **Only text generated from non-overlapping image data** enters ReText.
>
> 3. It is unclear how textual descriptions alone, generated for images that do _not_ overlap with the evaluation datasets, could constitute a form of data leakage for image-based person Re-ID.
>
> 4. **If a leak existed, performance should inflate only on overlapped datasets — but this is not observed.** Our method achieves strong generalization even on datasets completely absent from the captioner training, e.g., CUHK-SYSU (see Table 6). If leakage were present and artificially boosting performance, we would not expect such consistent improvements on datasets with **absolutely zero overlap.**
>
> _Summary_
>
> - There is _no_ image-level overlap.
>
> - The captioner is _not_ used in ReText; only generated text for non-overlapping images is used.
>
> - Strong performance on entirely disjoint datasets further confirms the absence of leakage.
>
> We hope this detailed clarification resolves the concern.

---

### Official Review · Reviewer_pJnk · 2025-10-31

**Soundness:** 2
**Presentation:** 3
**Contribution:** 2
**Rating:** 4
**Confidence:** 4

**Summary:**

The paper proposes ReText, a method that enhances cross-domain generalization in Re-ID by training on a mix of multi-camera Re-ID data and single-camera data with textual descriptions. It jointly optimizes Re-ID, image-text matching, and text-guided image reconstruction, and achieves favorable experimental results.

**Strengths:**

1. The figures and tables of this paper are relatively clear, and its writing is easy to understand.

2. It uses mixed training of multi-camera Re-ID data and text-annotated single-camera data, addressing both the scarcity of multi-camera data and the lack of cross-view variation in single-camera data.

3. ReText achieves better experimental results on multiple datasets compared with multiple existing methods.

**Weaknesses:**

1. Multi-task joint learning is not a novel concept in fields such as person re-identification or person retrieval. Many works employ methods based on generation/reconstruction and image-text matching for joint learning with ReID. The methods used in each task in this paper are mostly the introduction or adaptation of existing technologies; thus, the improvements in framework design are incremental.

2. The authors note in the implementation details that ReText is trained on a hybrid dataset constructed from multi-camera data and single-camera data. This is in conflict with the settings (Protocol 1) presented in Table 5. If ReText is trained using mixed data while other methods adhere to the settings of Protocol 1, such a comparison clearly lacks fairness—we cannot determine whether the performance gain stems from the model itself or the large-scale data.

3. ReMix is the most relevant work to this paper. However, existing experiments only show the comparison between ReText and ReMix under their respective settings, and lack an intuitive and comprehensive comparison. For example, a comparison between ReText and ReMix based on the same data and settings.

4. By combining Table 3 and Table 5, we note that ReText, without introducing other tasks and only using a pure ReID model, has an accuracy that far exceeds all comparative methods. This is quite confusing, because ReText does not make any additional improvements to the ReID model. It adopts the same loss function as ReMix, yet the accuracy of its pure ReID model is 20% higher than that of the complete ReMix. The authors do not analyze the source of these gains.

5. ReID is an application-oriented task. Does the authors' adoption of multi-task joint learning increase the difficulty of actual training, as well as raise computational cost and model complexity? This paper lacks relevant analysis.

**Questions:**

Please refer to the Weaknesses.

---

> ### Author Response · Authors · 2025-11-19
> **Weakness 1**
>
> We thank the reviewer for the feedback. Below we address the concerns raised in Weakness 1.
>
> > _Multi-task joint learning is not a novel concept in fields such as person re-identification or person retrieval._
>
> While multi-task joint learning has indeed been explored in person re-identification, the primary objective of such methods (e.g., Instruct-ReID \[1] and Instruct-ReID++ \[2]) is to enable a **single model to handle multiple categories of Re-ID tasks simultaneously**: image-based person Re-ID, text-based person Re-ID, visible–infrared Re-ID, clothes-changing Re-ID, and others.
>
> In contrast, **our work focuses exclusively on improving generalization within the image-based person Re-ID task**.
>
> **Our work is the first to introduce training on a mixture of multi-camera and text-enriched single-camera data.** The only prior method that explored this paradigm is ReMix \[3], but it did **not** investigate the impact of single-camera data, which are inherently simple from a multi-camera Re-ID perspective. In ReText, we **increase the complexity of single-camera data using textual descriptions** and show that this substantially improves generalization (Section 4.3, Proof-of-Concept).
>
>
> > _Many works employ methods based on generation/reconstruction and image-text matching for joint learning with ReID._
>
> To the best of our knowledge, **there are no existing works that use generation/reconstruction and image-text matching in combination with Re-ID for the purpose of studying generalization in image-based person Re-ID**, nor works that jointly train on a mixture of multi-camera and single-camera data with text under this setting.
>
>
> > _The methods used in each task in this paper are mostly the introduction or adaptation of existing technologies._
>
> In ReText, we introduce three tasks: Re-ID, image–text matching, and text-guided image reconstruction. Each task has a specific motivation:
>
> **-** Re-ID task: we study the effect of incorporating simple single-camera data and **show that the ReMix strategy of computing the Re-ID loss jointly for multi-camera and single-camera data is not optimal** (Section A.3, Using Single-camera Data in Re-ID Losses).
>
> \- Image–text matching task: we propose two new loss functions — Identity-aware Matching Loss and Structure-preserving Loss — and **demonstrate that they improve image-text alignment in the context of image-based person Re-ID** (Section 4.4.2, Effect of Matching Loss Design).
>
> \- Image reconstruction guided by text: we introduce a reconstruction approach that, in addition to unmasked patches, **incorporates textual descriptions to guide the reconstruction process unlike MAE \[4],** which relies solely on visual patches.
>
> \[1] He, W., Deng, Y., Tang, S., Chen, Q., Xie, Q., Wang, Y., ... & Yan, Y. (2024). Instruct-reid: A multi-purpose person re-identification task with instructions. In Proceedings of the IEEE/CVF Conference on Computer Vision and Pattern Recognition (pp. 17521-17531).
>
> \[2] He, W., Deng, Y., Yan, Y., Zhu, F., Wang, Y., Bai, L., ... & Tang, S. (2025). Instruct-reid++: Towards universal purpose instruction-guided person re-identification. _IEEE Transactions on Pattern Analysis and Machine Intelligence_.
>
> \[3] Mamedov, Timur, Anton Konushin, and Vadim Konushin. "ReMix: Training generalized person re-identification on a mixture of data." 2025 IEEE/CVF Winter Conference on Applications of Computer Vision (WACV). IEEE Computer Society, 2025.
>
> \[4] He, K., Chen, X., Xie, S., Li, Y., Dollár, P., & Girshick, R. (2022). Masked autoencoders are scalable vision learners. In Proceedings of the IEEE/CVF conference on computer vision and pattern recognition (pp. 16000-16009).

---

> ### Author Response · Authors · 2025-11-19
> **Weakness 2**
>
> _Concerns about the honesty of the comparison (weakness 2)._
>
> We appreciate the reviewer’s concern regarding fairness in comparison. In our case, **the use of additional single-camera data and its joint interaction with multi-camera data is not an auxiliary option but the core mechanism** of ReText itself.
>
> As we said in Section A.7.3 Remark 3, only ReMix and our ReText are trained using additional single-camera data, which is central to the design of both methods. All other approaches rely solely on labeled multi-camera datasets, as they are not created to handle heterogeneous training data. **Therefore, adapting them to use single-camera data would lead to an unfair degradation of their performance. For this reason, their results are reported under the official training settings.** Moreover, ReText is the first method that uses textual description for simple single-camera data by design.
>
> **We must highlight that our experiments show that ReMix is less effective at leveraging single-camera data compared to our ReText**, and we also demonstrated that the idea of using  textual description for simple single-camera data is wealthy (Section 4.3, Proof-of-Concept).
>
> At the same time, we fully acknowledge the importance of fair comparison and would welcome other ideas from the reviewer on what evaluation strategy would be considered most appropriate in this scenario.

---

> ### Author Response · Authors · 2025-11-19
> **Weakness 5**
>
> _Concerns about computational cost and model complexity (weakness 5)._
>
> Thanks for the concerns. As stated in line 147, only the image encoder is used at inference time, which means that the proposed multimodal joint learning strategy operates exclusively during training. Consequently, **they do not increase inference complexity, computational cost, or deployment requirements.**

---

> ### Author Response · Authors · 2025-11-19
> **Weaknesses 3 and 4**
>
> _Fair comparison with ReMix (weaknesses 3 and 4)._
>
> We would like to thank the reviewer once again for the fair and constructive comments.
>
> Regarding the suggestion to provide a comparison between ReText and ReMix under the same data and settings, unfortunately **this is not feasible for the following reasons:**
>
> 1. _ReText and ReMix rely on fundamentally different types of single-camera data._ ReText uses single-camera data from **SYNTH-PEDES**, whereas ReMix uses **LUPerson**. ReMix relies on a specific annotation and sampling strategy that **requires the video identifier for each image**, which is available in LUPerson but **not** in SYNTH-PEDES. Therefore, **ReMix cannot be trained using SYNTH-PEDES single-camera data.** Conversely, **LUPerson does not contain textual descriptions**, which are essential for training ReText. **As a result, it is inherently impossible to train both methods on the same single-camera dataset.**
>
> 2. _ReText and ReMix use different architectures, and unifying them is not possible._ ReText is built around a **Vision Transformer**, which is integral to the core idea of our method. ReMix, however, is based on **ResNet50-IBN**. **We cannot replace the backbone in ReText without breaking the design of our method**, and we also cannot re-train ReMix with a ViT backbone because its official code is **not publicly available**.
>
> Despite these constraints, we want to emphasize that this does not undermine the validity of our empirical evaluation of ReText:
>
> 1. _SYNTH-PEDES and LUPerson are comparable in scale._ Both datasets contain on the order of 4 million person images harvested from YouTube videos. Thus, **differences in dataset size cannot explain the performance gains of ReText over ReMix.**
>
> 2. _Table 2 (Section 4.3, Proof-of-Concept) provides an indirect yet fair comparison of ReText and ReMix._ Rows 4 and 5 of Table 2 effectively compare:
>
>    - a reproduction of ReMix under our unified training setup (row 4) — in this setting we also use a mixture of multi-camera and single-camera data as it was proposed in ReMix, and
>
>    - ReText trained on the same data with the same backbone, but with text used to enrich single-camera data (row 5).
>
> **The results clearly demonstrate that enriching single-camera data with textual information — our main idea — provides significant improvements in comparison with ReMix. This validates the conceptual contribution of ReText, even when controlling for backbone and training data.**
>
> Furthermore, **Tables 5 and 6 include several ViT-based methods, and ReText outperforms all of them**, reinforcing that our improvements come from the proposed learning strategy rather than from architectural differences or specific settings.
>
> Regarding the 20% performance gap between ReText and ReMix, as we noted both above and in the paper itself: **ReText outperforms ReMix when trained using only Re-ID losses primarily because it employs a more powerful image encoder** (ReText uses a Vision Transformer, whereas ReMix relies on ResNet50-IBN). This architectural difference naturally leads to stronger baseline performance.
>
> However, we would like to reiterate that the comparison remains fair. If we look at rows 4 and 5 of Table 2 in Section 4.3 (Proof-of-Concept), we observe that **our method with textual supervision (row 5) clearly outperforms our own implementation of ReMix without text (row 4), under exactly the same training setup and using the same backbone.** This demonstrates that the improvement comes from the proposed idea of enriching single-camera data with textual information — rather than from encoder differences — thus confirming the fairness and validity of our comparison.

---

### Official Review · Reviewer_jZSV · 2025-11-01

**Soundness:** 3
**Presentation:** 2
**Contribution:** 2
**Rating:** 4
**Confidence:** 4

**Summary:**

The paper focuses on image-based person ReID, and proposes ReText, which utilizes both multi-camera ReID and single-camera ReID datasets for generalization. Specifically, ReText includes three losses: ReID Loss on multi-camera data, image-text matching loss on single-camera data, and image reconstruction loss on single-camera data. To further improve the performance, this paper also refines the original CLIP image-text matching loss into Identity-aware Matching loss (by softening) and a Structure-preserving loss to preserve the semantic during image-text matching. Experiments are conducted by combining several multi-camrea datasets and one single-camera dataset (SYNTH-PEDES). Ablations show the effectiveness of the proposed modules.

**Strengths:**

The description of the proposed method is easy to follow.

**Weaknesses:**

1. Lack of comparison with some important SOTA methods. (1) Given that the proposed method is trained on a large-scale dataset (see Tab.1, more than 4.7M images), recent large-scale pretraining methods for ReID should be included during comparison. For example, PLIP[1], which only uses SYNTH-PEDES but shows a better performance on Market. (2) Besides, this paper is also related to multimodel multitask training, therefore, Instruct-ReID[2] should also be included, which also achieves great performance on Market.

2. Regarding the experimental section, the comparative descriptions are not clearly presented. It seems that the method is first pretrained on the large-scale dataset (Tab.1), then finetuned on different settings (Tab.5). But I can not confirm.

3. Overclaim on "first work to explore multimodal joint learning on a mixture of multi-camera and single-camera data in image-based personRe-ID".  "multimodal joint learning on a mixture of multi-camera and single-camera data" seems like a subset of multimodal multitask learning Re-ID (Two tasks: traditional image-based Re-ID, image-text Re-ID). There are some existing works like Instruct-ReID, Instruct-ReID++[3].

4. Small weakness. The improvement of the image reconstruction task is small, but it includes more computation costs.

[1] PLIP: Language-Image Pre-training for Person Representation Learning
[2] Instruct-ReID: A Multi-purpose Person Re-identification Task with Instructions
[3] Instruct-reid++: Towards universal purpose instruction-guided person re-identification

**Questions:**

Please refer to weaknesses. My concerns are mainly about setting (weakness 3), performance (weakness 1) of this paper. If the paper is truly pre-training on  a large-scale dataset (see Tab.1, more than 4.7M images), then finetuned, the shown performance is limited. Additionally, the ablations demonstrate the effectiveness of the proposed modules, allowing the community to have a better choice.

---

> ### Author Response · Authors · 2025-11-19
> **Overall comment**
>
> We thank the reviewer for the helpful and constructive feedback. Before addressing specific comments, we would like to restate the core idea of the proposed method to avoid possible misunderstandings.
>
> ReMix [1] was the first work to introduce mixing multi-camera and single-camera data during the fine-tuning stage — a completely new approach in image-based person Re-ID. In our method ReText, we take this idea further and demonstrate that **ReMix has limited generalization** because single-camera data are inherently too simple for multi-camera person re-identification (see Section 4.3, Proof-of-Concept). To address this limitation, **ReText is the first to use text to increase the complexity of training on single-camera data within the image-based person Re-ID setting.** As a result, ReText becomes **the first method that jointly trains on multi-camera and single-camera data enriched with textual descriptions.**
>
> This is the central idea behind our approach. To achieve effective learning in this setting, we introduce a three-task training paradigm (Re-ID, image-text matching, and text-guided image reconstruction), together with novel loss functions (Identity-aware Matching Loss and Structure-preserving Loss). We therefore emphasize that **ReText is a fundamentally new solution for image-based person Re-ID**, and previously mentioned methods such as PLIP and Instruct-ReID do not overlap with our approach.
>
> Below we address your comments in detail.
>
> [1] Mamedov, Timur, Anton Konushin, and Vadim Konushin. "ReMix: Training generalized person re-identification on a mixture of data." 2025 IEEE/CVF Winter Conference on Applications of Computer Vision (WACV). IEEE Computer Society, 2025.

---

> ### Author Response · Authors · 2025-11-19
> **Weaknesses 1 and 2**
>
> _Lack of comparison with some important SOTA methods and concerns about fine-tuning (weaknesses 1 and 2)._
>
> Thanks for the reviewer’s observation. In Table 5, the Training datasets column denotes a two-dataset configuration: a multi-camera dataset paired with the single-camera dataset SYNTH-PEDES. This pairing is used consistently across all experiments. Instead of pre-training ReText on a large single-camera dataset and subsequently fine-tuning on a multi-camera dataset, our framework performs joint training on the corresponding pair. Between experimental blocks in Table 5, only the multi-camera component of the pair changes.
>
> **Our paper and experiments primarily focus on the cross-domain scenario** for demonstration of the method’s generalization (i.e., training on one multi-camera dataset and evaluating on another). Accordingly, we did not include methods whose papers do not report results in comparable cross-domain settings. Nevertheless, Table 5 additionally contains in-domain results for the training pairs (Market-1501, SYNTH-PEDES) and (MSMT17, SYNTH-PEDES) evaluated on their respective test splits.
>
> We agree that including PLIP and Instruct-ReID under the same in-domain setup would strengthen the comparison. **Importantly, ReText outperforms both methods under the corresponding training conditions:**
>
> \- PLIP pre-trained on SYNTH-PEDES and fine-tuned on Market-1501 reports 93.2 mAP on Market-1501.
>
> \- Instruct-ReID with large-scale pre-training fine-tuned on Market-1501 reports 93.5 mAP on Market-1501; fine-tuned on MSMT17 it reports 72.4 mAP on MSMT17.
>
> \- ReText, trained simultaneously on SYNTH-PEDES and Market-1501, achieves 93.8 mAP on Market-1501; trained on SYNTH-PEDES and MSMT17, it achieves 83.6 mAP on MSMT17.
>
> These results confirm that ReText maintains state-of-the-art or superior performance even when compared to recent approaches trained on the large-scale dataset. **This fact demonstrates the effectiveness of our strategy of joint training on multi-camera and single-camera data enriched with textual descriptions.**

---

> ### Author Response · Authors · 2025-11-19
> **Weakness 3**
>
> _Concerns about claim "first work to explore multimodal joint learning on a mixture of multi-camera and single-camera data in image-based person Re-ID" (weakness 3)._
>
> We thank the reviewer for this valuable comment. As stated earlier, **ReMix and our ReText are the first methods to adopt a joint training strategy on both multi-camera and single-camera data**. To the best of our knowledge, **no prior work has explored how textual supervision can enrich single-camera data to improve cross-domain transferability in image-based person Re-ID**. When we refer to “multimodality,” we specifically mean **the combination of multi-camera and single-camera data during training, as well as the use of text to make single-camera data harder for the person Re-ID task**.
>
> The works mentioned by the reviewer, Instruct-ReID and Instruct-ReID++, belong to a different class of methods: they are designed to simultaneously address multiple categories of person re-identification tasks (image-based Re-ID, text-based Re-ID, visible–infrared Re-ID, clothes-changing Re-ID, etc.). This is fundamentally different from our focus. In contrast, **our work concentrates exclusively on the image-based person Re-ID task**, and we show that generalization across domains can be improved through a combination of heterogeneous visual data and textual supervision.
>
> We agree that phrases such as _“first work to explore multimodal joint learning on a mixture of multi-camera and single-camera data in image-based person Re-ID”_ and _“multimodal joint learning on a mixture of multi-camera and single-camera data”_ may be potentially confusing for readers. We would be very grateful if the reviewer could suggest alternative wording that they find clearer. We are fully prepared to revise the positioning either during the discussion phase or in the camera-ready version.

---

> ### Author Response · Authors · 2025-11-19
> **Weakness 4**
>
> _Small weakness. The improvement of the image reconstruction task is small, but it includes more computation costs (weakness 4)._
>
> We thank the reviewer for this comment. We agree that the contribution of the Image Reconstruction Task is smaller compared to the other components. However, we would like to emphasize that incorporating this task during training **does not increase the computational cost of ReText at inference time**. As stated at the beginning of Section 3 (Proposed Method), **only the momentum image encoder is used during inference**.
>
> We also emphasise that achieving high-fidelity image reconstruction was _not_ our goal in this work (although the qualitative results in Figure 3 show that the reconstructions are reasonably good). Instead, our aim was to demonstrate **how textual information can be leveraged as an additional modality during training** for the image-based person Re-ID task. We believe that ReText can serve as a foundation for future research in this direction.

---

> ### Comment · Reviewer_jZSV · 2025-11-27
>
> Thank you.
>
> After reading the rebuttal, I still have some questions.
>
> 1. The statement in your response to W3, 'ReMix and our ReText are the first methods to adopt a joint training strategy on both multi-camera and single-camera data', is **not true**. As I mentioned before, Instructed-ReID also used a joint training strategy on multi-camera (Traditional ReID), single-camera data (Language-Instructed ReID) and other ReID tasks. By the way, after re-reading the Instructed-ReID paper, it is also a joint-training multimodal ReID model and does not need finetuning (response to W2), which is similar to the evaluation protocol stated by the authors.
>
> 2. In response to W1 and 2,  "trained on SYNTH-PEDES and MSMT17, it achieves 83.6 mAP on MSMT17"? Why a performance improvement from the original submission?

---

> > ### Author Response · Authors · 2025-11-28
> >
> > Thank you for your follow-up and for carefully reading our rebuttal. We appreciate the opportunity to further clarify these points.
> >
> > **1. Regarding joint training and the relation to Instruct-ReID**
> >
> > You are correct that Instruct-ReID adopts joint training over multiple Re-ID tasks. However, our claim is specific to a different problem setting and research question. Instruct-ReID demonstrates that a single model can simultaneously handle multiple types of Re-ID tasks (image-based, text-based, visible–infrared, clothes-changing, etc.). **It does not investigate how incorporating additional data sources affects the performance of a single target task, nor does it study generalization ability.**
> >
> > In particular:
> > - Instruct-ReID evaluates performance in a **within-dataset setting**, where training and testing are conducted on different splits of the same datasets (e.g., training on MSMT17 training set + auxiliary datasets and testing on MSMT17 test set).
> > - **The effect of single-camera data on multi-camera image-based Re-ID generalization is not analyzed.**
> > - **Cross-dataset evaluation — where the training and testing domains differ — is not considered in Instruct-ReID.**
> >
> > In contrast, ReText focuses exclusively on image-based person Re-ID and studies a different question: how to improve cross-dataset generalization by jointly training on heterogeneous data sources.
> >
> > In this context, ReMix and our ReText are the first works to show that stylistically diverse single-camera data can improve generalization, and that this improvement requires careful treatment — since naive inclusion of such data actually degrades cross-dataset performance, as shown both in ReMix and in our own analysis.
> >
> > **2. Regarding the reported MSMT17 performance**
> >
> > Thank you for pointing this out, and we apologize for the confusion. This was a typographical error in our previous response.
> >
> > The correct statement is:
> >
> > ReText, trained jointly on SYNTH-PEDES and Market-1501, achieves 93.8 mAP on Market-1501; trained on SYNTH-PEDES and MSMT17, it achieves 78.7 mAP on MSMT17.
> >
> > This correction does not affect our conclusions: ReText still outperforms the corresponding baselines.
> >
> > We hope these clarifications address your concerns, and we thank you again for the thoughtful discussion.

---

### Author Response · Authors · 2025-11-19
**Authors' answer**

We would like to thank all reviewers for their valuable comments, questions, and suggestions. We have addressed all raised points in detail and would greatly appreciate any further feedback that could help us improve the paper. We hope that, with the reviewers’ input, we will be able to refine and strengthen our work for the camera-ready version.

---

### Author Response · Authors · 2025-11-20

Dear Reviewers,

We have updated the manuscript to incorporate your valuable feedback. The following changes have been made:
- Section 2.4 “Difference from Existing Multimodal Person Re-ID Methods” has been added to better clarify the core idea of our work and how ReText differs from existing approaches.
- Line 191: We now explicitly state that all proposed tasks contribute equally to the overall training objective.
- Lines 344–346: We added a clarification regarding the absence of data leakage between multi-camera and single-camera datasets.
- Lines 392–396: We included additional conclusions following the Proof-of-Concept experiments to reinforce the main motivation and insights of our work.
- Lines 430–431: A footnote has been added to explain the fairness of our comparison with ReMix.
- Lines 522–529: We added further discussion highlighting the fairness of our experimental comparison.
- Table 6 has been moved to the main paper.

We hope these updates address your concerns and help improve the clarity and quality of the paper.

Sincerely,

The Authors

---

### Meta-Review · Area_Chair_aG3m · 2026-01-06

**Summary:**

This paper aims to improve generalizable person re-id for the image modality only. The method is to jointly train on multi-camera re-id data and single-camera re-id data with text descriptions. The model is trained with three objectives. The results look quite promising.

Reviewers, however, have many questions and gave 4,4,0,4 ratings. I think the biggest concern is novelty / that claims are not well justified.   Reviewers think the claims and methods have been studied before and that 'first' may sound over-selling. Authors provided rebuttal to address this concen which has been read by the AC. AC thinks that authors' rebuttal make some sense in differentiating from existing works. However, the issue is that the paper is written in a way that states novelty in all aspects. For example, abstract reads "We propose ReText, a novel method trained on a mixture of multi-camera Re-ID data and single-camera data, where the latter is complemented by textual descriptions to enrich semantic cues. During training, ReText jointly optimizes three tasks: (1) Re-ID on multi-camera data, (2) image-text matching, and (3) image reconstruction guided by text on single-camera data."

AC advises the authors to tune down the novelty a bit and instead acknowledge more the contributions from existing works. Authors could try to dig deeper into issues of not using text descriptions in ReMix. This would make this paper more focused and solve the concerns from the reviewers.

AC recommends reject and encourages authors to make revisions to this work.

**Reviewer Concerns:**

Novelty and over-statement are the main concerns.

**Reviewer Scores:**

It's unlikely reviewers will raise their scores.

---

### Decision · Program_Chairs · 2026-01-26

Reject